# Use of *Pseudomonas protegens* to Control Root Rot Disease Caused by *Boeremia exigua* var. *exigua* in Industrial Chicory (*Cichorium intybus* var. *sativum* Bisch.)

**DOI:** 10.3390/plants13020263

**Published:** 2024-01-17

**Authors:** Tamara Quezada-D’Angelo, Juan San Martín, Braulio Ruiz, Pía Oyarzúa, Marisol Vargas, Susana Fischer, Pamela Cortés, Patricio Astete, Ernesto Moya-Elizondo

**Affiliations:** 1Departamento de Producción Vegetal, Facultad de Agronomía, Universidad de Concepción, Chillán, Chile; tamaraquezada@udec.cl (T.Q.-D.); juasanmartinm@udec.cl (J.S.M.); braruiz@udec.cl (B.R.); piaoyarzua@udec.cl (P.O.); marisolvargas@udec.cl (M.V.); sfischer@udec.cl (S.F.); macortes@udec.cl (P.C.); 2Departamento de Investigación y Desarrollo, Orafti-Beneo S.A., Pemuco, Chile

**Keywords:** root rots, chicory, 2,4-diacetylphloroglucinol, inulin

## Abstract

*Boeremia exigua* var. *exigua* is a recurrent pathogen causing root rot in industrial chicory. Currently, there is no chemical or varietal control for this disease, and thus, management strategies need to be developed. This study determined the biocontrol effect of strains of *Pseudomonas protegens* bacteria with antimicrobial compounds on the fungus *B. exigua* var. *exigua* under in vitro, in vivo, and field conditions. In addition, root colonization by these bacteria was estimated by the *phlD*-specific PCR-based dilution end point assay. Eighteen isolates of *Pseudomonas* spp were evaluated, and the strains that showed the greatest in vitro inhibition of fungal mycelial growth (mm), Ca10A and ChB7, were selected. Inoculation with the strain ChB7 showed less severity (necrotic area) under in vivo conditions (root trials) compared with the control inoculated with the pathogen (*p* ≤ 0.05). The molecular analysis revealed that the root colonization of plants grown in pots was equal to or greater than 70%. Similar levels were observed in the field trials conducted at the Selva Negra and Canteras experimental stations (2015–2016 season), with values ranging from 85.7 to 70.5% and from 75.0 to 79.5%, respectively. Regarding yield (ton ha^−1^), values were higher in the treatments inoculated with strains Ca10A and ChB7 (*p* ≤ 0.05) at both experimental sites, while a lower incidence and severity of root rot were observed at Selva Negra. These results suggest that the Chilean strains of *P. protegens* are a promising tool for the control of root diseases in industrial chicory.

## 1. Introduction

Chicory (*Cichorium intybus* var. *sativum* Bisch.) is industrially cultivated in Chile for the extraction of inulin and oligofructose from the roots [1]. Inulin is a fructan polymer, which is used as a functional ingredient in the food industry because of its prebiotic properties that improve human health and reduce the risks of developing several diseases [2]. Chicory was introduced to Chile in 2006, being cultivated mainly in the Biobío Region (36°46′22″ S, 73°03′47″ W), with exports of 13,000 tons per year for approximately USD 50 million FOB, which has ranked Chile as the third largest exporter of inulin worldwide [3]. However, this industrial crop is affected by the ascomycete fungus *Boeremia exigua* var. *exigua* (Desm.) Aveskamp, Gruyter & Verkley, which is one of the pathogens associated with the complex of fungi and pseudo-fungi causing “root rot” [4]. In fact, this disease is an important yield-limiting factor in chicory production in the southern area of Chile, resulting in severe decreases in annual crop yields ranging from 3 to 80%.

*Boeremia exigua* var. *exigua* can parasitize many plant species, but it can also grow on dead plant tissue [5,6]. The fungus acts as an opportunistic parasite that can cause pre- and post-harvest root rot, necrosis, and lesions of leaves and stems in various host plant species [5,7,8,9]. In industrial chicory, *B. exigua* var. *exigua* causes dark brown to black necrotic lesions with a dry and firm sunken consistency, mainly located in the upper part of the roots and characterized by a sharp boundary between diseased and healthy tissue [4]. Currently, there is no chemical or varietal control of this pathogen in industrial chicory [9,10], while there is also a lack of technical knowledge to manage the disease. However, risks of root rot in industrial chicory can be minimized by using management practices, such as proper soil selection and adequate irrigation water supply, and by preventing excess moisture in the soil during the summer, particularly if temperatures exceed 20 °C [10] since these conditions favor pathogen attack and disease development in the crop. In fact, the optimum growth temperature for *B. exigua* var. *exigua* is between 20 and 25 °C, which implies that strategies for root rot control in industrial chicory need to be developed and implemented.

Interest in the biological control of crop diseases has increased in recent years due to increasing public concern about the impacts of chemical pesticides on the environment, human health, and food safety [11,12,13,14,15]. In this sense, the use of microorganisms capable of controlling soil pathogens constitutes an environmentally friendly alternative for the control of industrial chicory root rot.

Bacteria of the *Pseudomonas fluorescens* group have been intensively studied for disease control in agriculture [16]. These rhizobacteria are mostly isolated from plant roots, contributing to the suppression or reduction in root diseases caused by different bacteria, fungi, pseudo-fungi, and phytopathogenic nematodes in different crops [11,17,18,19]. The protective mechanisms of these bacteria include induced systemic resistance, competition, and antibiosis associated with the production of antimicrobial secondary metabolites, such as 2,4-diacetylphloroglucinol (2,4-DAPG), pioluterine, pyrrolnitrin, phenazines, or hydrogen cyanide [17,18]. It has been indicated that bacteria capable of producing 2,4 DAPG (*phlD*+) and pioluterin (*plt*+) are molecularly and phenotypically different from other strains of *P. fluorescens* and thus considered as a new species called *Pseudomonas protegens* [17]. The presence of 2,4-DAPG is associated with toxicity to a broad spectrum of organisms, such as bacteria, fungi, and nematodes [20], while pioluterin is mainly known for its suppressive capacity against the oomycete *Pythium ultimum* [21,22], as well as its antibacterial activity [23]. The antifungal activity of 2,4-DAPG is characterized by the dissipation of the proton gradient across the mitochondrial membrane, affecting cellular respiration in pathogenic fungi [24] and causing cell lysis in bacteria [20].

To the best of the authors’ knowledge, there is no evidence of the ability of *P. protegens* to reduce the attack of *B. exigua* var. *exigua* in industrial chicory. Therefore, the objectives of this research were to select *Pseudomonas protegens* strains and determine their effects on the pathogenic fungus *Boeremia exigua* var. *exigua* under in vitro, in vivo, and field conditions. Additionally, the root colonization capacity of these bacteria was also evaluated on seed-treated plants of industrial chicory by using the *phlD*-specific PCR-based dilution end point assay.

## 2. Results

### 2.1. In Vitro Preselection of the Antagonistic Effect of Pseudomonas protegens Strains on the Pathogenic Fungus B. exigua var. exigua under In Vitro Conditions

The cluster analysis allowed determining three groups of bacteria associated with mycelial growth inhibition of *B. exigua* var. *exigua* (Figure 1). The first cluster, strains C21BD1, C12BC1, C21BC3, C9BA1, and Ca10B, presented the lowest GII (%), with an average value of 1.2%; the second cluster, strains C5BE4, Ca9B, and CA1A, reached an average value of 25.7% GII. Finally, the third cluster, strains Ca11A, Ca12A, Ca10A, ChC7A, Ca8A, Ca4A, Ca3B, Ca5A, Ch8B8, and ChB7, caused the greatest inhibition of mycelial growth, reaching an average value of 53.9% GII (Figure 1).

To perform the dual culture assays, the strains of *P. protegens* exhibiting the highest inhibition of mycelial growth (third cluster), ChB7 (54.7%), Ch8B8 (54.5%), and Ca5A (54.4%), were selected. In addition, the strain Ca10A (51.7%) was also selected because it showed high antagonism against another pathogen causing root rot (Figure 1).

The results of the dual culture assays showed that the Chilean strains inhibited mycelial growth of the pathogen *B. exigua* var. *exigua* after 14 days of incubation, with GII values ranging from 30 to 44% (Figure 2). Strains ChB7 and Ca10A were responsible for the greatest growth inhibition of the pathogen in dual cultures, reaching 44 and 36.7% GII, respectively (Figure 2), and were selected for subsequent evaluations.

### 2.2. Evaluation of the Control Effectiveness of Pseudomonas protegens on Boeremia exigua var. exigua in Roots of Industrial Chicory under Controlled Conditions

Differences in rot damage index (*p* ≤ 0.001) were observed between roots inoculated (*n* = 8 roots per treatment) with the antagonistic bacteria and the fungal pathogen after 21 days of incubation in a humid chamber. The non-inoculated control did not present decay in the injured area, being different from the rest of the treatments. Regarding the treatments inoculated with the fungus, ChB7 was statistically different, with a damaged zone being 8.5 mm smaller than that observed in the control only inoculated with *B. exigua* var. *exigua* strain Pho669. Furthermore, the treatment with the strain Ca10A was not different from the control inoculated with Pho669 or from the treatment with ChB7 (Figure 3).

### 2.3. Evaluation of the Root Colonization Capacity and Control Effectiveness of Antagonistic Bacteria (P. protegens) on Root Rot Caused by the Pathogen B. exigua var. exigua in Roots of Industrial Chicory under Controlled and Field Conditions

Based on the proportion of positive samples for the *phID* gene, colonization of strain Ca10A was equal to or greater than 80% (*n* = 10 plants) in the evaluations at 30, 90, and 180 days after bacterial inoculation. In the case of the plants inoculated with the strain ChB7, it was found that over 70% of the roots tested positive for the *phID* gene during the cultivation period, while 10 roots were positive for the presence of the indicator gene after inoculation (100%). Even though the experiment was conducted in pots under controlled conditions and the substrate was autoclaved, three root systems of the 10 control plants (without inoculation) tested positive for the *phID* gene (30%) at 30 days after bacterial inoculation, but this was observed in only one root (10%) at 180 days after bacterial inoculation (Figure 4).

No significant differences were observed between the plants inoculated with *P. protegens* and the non-inoculated control in terms of the fresh and dry weights of leaves and roots.

In the first semester of crop development under field conditions, with evaluations at 30, 60, and 90 days after seeding (DAS), it was observed that the presence of the *phlD* gene varied between 85.7% and 67.9% and between 75% and 53.6% at Selva Negra and Canteras, respectively (Figure 5). In the second semester, which included evaluations at 150 DAS and harvest, colonization by populations of *P. protegens* bacteria with the *phlD* gene varied between 82.1% and 70.5% and between 71.4% and 79.5% at Selva Negra and Canteras, respectively (Figure 5). No differences were found between the two experimental stations in the detection of bacteria with the presence of the *phlD* gene in the analyzed roots (t-value = 0.71, *p* = 0.2078).

The average values obtained at harvest reached 113.3 and 65.4 ton ha^−1^ of chicory at Selva Negra and Canteras, respectively. With respect to leaf weight, significant differences were found between the treatments in both experiments (*p* ≤ 0.05; Table 1 and Table 2). The treatment that mixed both bacterial strains produced 23.3% more leaf biomass than the non-inoculated control. However, no differences were observed when compared with the control inoculated with the pathogen at Canteras. In the experiment conducted at Selva Negra, a higher production of leaf biomass was observed for the treatment inoculated with both strains, being 20% higher than the control inoculated with the fungus, but it did not differ from the non-inoculated control (Table 1).

With respect to the weight of healthy roots (ton ha^−1^), significant differences were found between the treatments evaluated in the experiment conducted at the Selva Negra experimental station (*p* = 0.021). The seed treatment inoculated with strains Ca10A and ChB7 recorded a value of 111.9 ton ha^−1^ of healthy roots, which was significantly different from the 93.4 and 98.6 ton ha^−1^ obtained with the control inoculated with the fungus and the non-inoculated control, respectively (Table 1). Regarding the weight of diseased roots (ton ha^−1^), there were significant differences between the treatments (*p* = 0.005); the seed treatment inoculated with strains ChB7 and Ca10A, with or without soil inoculation with *B. exigua* var. *exigua*, and that inoculated with both bacterial strains recorded an average weight of diseased roots that was 47.2% lower than the control inoculated with the fungus (Table 1).

The results for the yield at harvest at the Canteras experimental station revealed that the treatments inoculated with the bacterial strain ChB7 and those inoculated with strains ChB7 and Ca10A reached values of 64.8 and 66.2 ton ha^−1^ of healthy roots, respectively, being 21% and 24.4% higher than the yield of 53.5 ton ha^−1^ recorded by the control inoculated with the fungus (*p* = 0.043) (Table 2). In addition, the leaf weight and diseased root weight (ton ha^−1^) showed significant differences between the treatments (*p* ≤ 0.05). However, these results were not relevant for the present investigation because the treatments inoculated with the antagonistic bacteria were significantly different between them, but no differences were found between these treatments and the control inoculated with the fungus (Table 2).

The incidence of root rot in the Selva Negra experiment was significantly lower in all the treatments compared with the control inoculated with *B. exigua* var. *exigua* and the non-inoculated control (*p* = 0.033). Similar results were obtained in terms of disease severity; five treatments were significantly lower than the control inoculated with the fungus (*p* ≤ 0.002; Table 3). In the Canteras experiment, most of the treatments showed no differences in terms of the incidence and severity rate of root rot, except for the treatment inoculated with the bacterial strain ChB7 and the fungal strain Pho669, which recorded the highest disease incidence (43.1%) and severity (23.5%) (Table 3). 

## 3. Discussion

The 2,4-DAPG-producing *Pseudomonas* spp. have been isolated from numerous crops such as beans, alfalfa, lentils, barley, oats, and wheat [19]. In the south of Chile, 2,4-DAPG-producing bacteria have been found in wheat crops and found responsible for the suppression of take-all disease caused by the fungus *Gaumannomyces tritici*, a phenomenon known worldwide as take-all decline, induced by monoculture of wheat [25,26,27]. However, there are no previous reports on the presence and biocontrol activity of these bacteria in industrial chicory.

The results obtained in the preselection of bacteria for the control of *B. exigua* var. *exigua* carried out in vitro and under controlled conditions in chicory roots suggested that the inhibitory effect induced by the bacterial strains would be due to the diffusion of the antimycotic compounds that have been described for *P. protegens* bacteria in the culture medium present in the Petri dish, such as 2,4-DAPG and pioluteorina. This occurs because this bacterial species is characterized by the presence of the *phlD* gene (determined in the bacteria used) and *plt* genes, which are associated with the production of the aforementioned compounds and contribute to the antagonism of *P. protegens* against different kinds of bacteria, fungi, and oomycetes [23]. To our knowledge, the current study is the first report of the inhibitory effect exerted by *P. protegens* bacteria on the fungus *B. exigua* var. *exigua*. The compounds influence the colonization of fungal tissues [28], but the sensitivity varies depending on the microorganism that is affected, e.g., pyoluteorin is twenty times more toxic than 2,4-DAPG for the growth of the bacterium *Erwinia amylovora* [29]. In addition, these two compounds also have different ecological roles. For example, 2,4-DAPG is toxic to the amoeba *Acanthamoeba castellanii* but not to pyoluteorin [30]. This implies that depending on the conditions faced by the bacterium, the production of one compound could be more advantageous in comparison with another [29]. For this reason, we did not determine which of the two antibiotics, or if both, were toxic to the pathogen *B. exigua* var. *exigua*. However, the antifungal activity of 2,4-DAPG, which is given by its ability to dissipate the proton gradient that occurs in the mitochondrial membrane, affecting fungal cellular respiration [24], suggests that this metabolite would be the main one responsible for controlling *Boeremia*. On the other hand, variability in root rot control in chicory could be explained by the fact that biosynthesis of antibiotics can be a metabolic burden for the bacterium. Although the biochemical and genetic determinants are not related to their biosynthesis, the production of 2,4-DAPG and pyoluteorin by *P. protegens* prevails one over the other in the face of competitors and predators or depending on the habitats of these bacterial populations [29].

Bacteria with antagonistic activity against soil-borne pathogens must establish and maintain population density in the rhizosphere environment [31]. Plants are the main determining factor of the microbial community structure because they provide carbon and energy sources to the soil microbiota through root exudates, forming a unique environment for bacterial colonization [32]. Our results indicated that the bacterial strains under study were capable of surviving in the chicory rhizosphere during crop development, which would indicate that root exudates released by industrial chicory plants favored the survival of *P. protegens*. Under field conditions, the evaluation of the presence or absence of *P. protegens* strains in chicory root indicated that the strains inoculated in the seed could increase in the chicory rhizosphere compared with the non-inoculated control. Determining the degree of bacterial colonization by seed inoculation was important given the capacity that the bacterial strains showed to control *B. exigua* var. *exigua* under controlled conditions, which may also account for the higher yields observed with their use in the field. As the used bacteria were originally isolated from wheat, it was important to determine their ability to survive and colonize the rhizosphere of industrial chicory, particularly considering that chicory plants have pivoting roots that can measure more than one meter in length and weigh up to 1000 g, whereas the wheat root system is fasciculate and thus is characterized by numerous thin roots.

In order for a microorganism to be beneficial and applicable to the crop, it is important that (i) it establishes or colonizes, (ii) it has the capacity to protect the plant, and (iii) it manages to compete with the native microbiota [33,34]. In the field experiments, there was a decrease in the percentage of positive samples for the *phID* gene in the evaluation at 90 DAS, while this was not observed in the pot trial. This difference could be explained by the fact that root growth rate increased up to 10 times at 90 DAS, tending to produce cracks in the tissue that released compounds rich in sugars; this could increase microbial populations and thus competition between them in the rhizosphere. In the case of the experiment with chicory plants under controlled conditions, the increase in root growth was not marked, possibly due to lower light and natural narrowness generated by the walls of the pots. In this sense, it is important to indicate that according to our observations in the field, rot symptoms started in this period. This agrees with [35], who reported that roots without wounds and inoculated with the pathogens *Phytophthora cryptogea* and *B. exigua* var. *exigua* did not show rot symptoms, indicating that the infections caused by these pathogens on the roots are dependent on the presence of wounds or lesions.

Higher percentages of bacterial samples positive for the *phID* gene were observed at Selva Negra compared with Canteras. This could be explained by the differences in soil type and soil properties between the experimental stations. According to [36], the soil at Selva Negra corresponds to a type of Andisol (“trumao”) soil, with high organic matter content ranging from 8 to 12% and excellent biological activity, while the soil at Canteras is sandy, with a low organic matter content, moderate biological activity, and excessive drainage. High drainage results in a rapid loss of water and nutrients applied to the crop, which may explain the lower yields observed at Canteras (Table 2). Nevertheless, colonization by bacteria of the *P. fluorescens* group depends on other factors or characteristics such as the pH of the rhizosphere, the temperature, the water flow in the soil profile (irrigation and rainfall), the plant species, and even the plant genotype [37]. In this study, not all of these variables were determined. However, differences between the experimental sites could have influenced the fact that Selva Negra recorded higher percentages of samples positive for the *phID* gene associated with populations of *P. protegens* in the roots. Likewise, the existence of populations of native bacteria of *P. protegens* naturally associated with the roots of industrial chicory were detected at both experimental sites, being higher at Selva Negra probably due to the higher content of organic matter in the soil.

There are few reports on the use of biological control agents of diseases that affect chicory cultivation in Chile [38]. Globally, there is also very little information on the use of these agents in this plant. Diseases such as root rot reduce production, which results in lower income for farmers who have contracts with Beneo-Orafti S.A. to grow industrial chicory since the company establishes acceptable quality levels for harvested chicory roots that show rot. In addition, the existence of roots damaged by root rot reduces the efficiency of inulin extraction and affects the safety of this product. Therefore, as a relevant finding of the present study, we would emphasize the increased yield of healthy roots and reduced root rot by the use of *P. protegens*, which was evidenced in the weight of diseased roots and incidence and severity of the disease (%) under field conditions. This improvement in yield variables and plant health occurred in the presence or absence of the fungal strain Pho669 of *B. exigua* var. *exigua*. It should be noted that the treatments inoculated with the mixture of the two Chilean strains of *P. protegens* obtained a higher yield in both experimental sites. Studies that have evaluated species of the *P. fluorescens* group isolated from suppressive soils and used as plant or soil inoculants [39] have demonstrated the efficiency of these bacteria to colonize roots, protect plants from different diseases, and increase plant productivity [40,41,42,43,44], also evidenced in our experience for industrial chicory.

The pathogen occurred naturally in the soil in which the study was conducted since the treatment without inoculation of *B. exigua* var. *exigua* presented roots with typical root rot symptoms. This was expected because the experimental sites were not fumigated prior to seeding, and those experimental sites were inoculated with this fungus in the past in cultivar resistance trials. In addition, a study conducted in Chile (Biobio Region) on seven crops of industrial chicory grown between San Carlos (36°25′29″ S, 71°57′29″ O) and Quilleco (37°28′00″ S, 71°58′00″ W) showed a high frequency of root rot symptoms of *B. exigua* var. *exigua* in all the sampled fields. Additionally, it is important to note that *B. exigua* var. *exigua* is characterized by being a facultative and opportunistic soil parasite that survives as a saprophyte in diseased or dead plant material through the formation of mycelium and/or pycnidia [9,45]. In fact, these adaptive conditions allow this pathogen to survive in the soil for long periods of time. For this reason, inoculation with the fungus in the experimental sites allowed standardizing the level of inoculum among the experimental plots.

The greater weight of healthy roots (ton ha^−1^) obtained with the seed treatments inoculated with the mixture of strains Ca10A and ChB7 at Canteras and Selva Negra, compared with the non-inoculated treatment and that inoculated with the fungus, may be associated with a growth-promoting effect when using both bacterial strains in the rhizosphere of chicory. The author of [46] evidenced the presence of growth-promoting metabolic mechanisms in strains Ca10A and ChB7, associated with phosphorous solubilizing activity and IAA (indole acetic acid) production. The author determined that strains Ca10A and ChB7 can solubilize 70% more phosphorus (mg of P_2_O_4_ L^−1^) than others of the genus *Pseudomonas* as reported by [33] and produce higher levels of IAA, which influences the hormonal balance of plants, affecting growth [47]. The ability to promote plant growth on industrial chicory and reduce the incidence and severity of root rot of Chilean strains of *P. protegens* is of commercial interest to the growers since industrial chicory is marketed for its pivoting root, and thus, an increase in the root growth and weight of healthy roots results in increased yield.

## 4. Materials and Methods

### 4.1. Isolates of B. exigua var. exigua

The highly pathogenic fungus *B. exigua* var. *exigua* strain Pho669, which was obtained from diseased roots of industrial chicory, was used in all the experiments. The high pathogenicity of this isolate was demonstrated through resistance trials conducted by Beneo-Orafti Chile S.A. in Chile by comparing different fungal isolates from different cultivars of industrial chicory. 

### 4.2. In Vitro Preselection of the Antagonistic Effect of Bacterial Isolates on the Pathogenic Fungus B. exigua var. exigua

A preselection of bacterial isolates with greater antibiosis against *B. exigua* var. *exigua* was conducted. Eighteen bacterial isolates, which were obtained from wheat plants [48], were evaluated. They corresponded to *Pseudomonas fluorescens* strains Ca1A, Ca9B, Ca11A, C21BD1, C12BC1, C21BC3, C9BA1, and C5BE4 and isolates of the bacterium *P. protegens* Ca3B, Ca4A, Ca5A, Ca8A, Ca10A, Ca10B, Ca12A, Ch8B8, ChC7A, and ChB7. All the isolates were obtained from the Phytopathology Laboratory of the Faculty of Agronomy, University of Concepción, Chile. Groups of six isolates were formed, and their inhibitory capacity against *B. exigua* var. *exigua* was compared. 

The bacterial isolates were previously incubated in King B (KB) broth (peptone protease, 20 g; anhydrous K_2_HPO_4_, 1.965 g; MgSO_4_ 7H2O, 1.5 g; 10 mL glycerol and 1000 mL water) at 25 ± 1 °C for 48 h with constant agitation at 150 rpm. An aliquot of 5 uL was placed equidistantly on a Petri dish with APD (20%) + KB medium, while a piece of isolated mycelium of the fungal strain Pho669 (5 mm in diameter), which was previously grown in APD medium at 24 °C for 10 days, was placed in the center of the dish. Petri dishes containing the fungus without bacteria were used as control. Prior to the experiments, the presence of the *phlD* gene, which is associated with the production of 2,4-DAPG, was determined in all the antagonistic strains under study. For this, the polymerase chain reaction (PCR) technique was used according to the protocol described by McSpadden et al. [49].

Measurements of the mycelial growth of *B. exigua* var. *exigua* were performed at 4, 7, 10, and 14 days of incubation by determining the distance between the antagonistic bacteria and the fungal hyphal edge. The growth inhibition index (GII) of the pathogen was determined and calculated once growth in the control (without bacteria) reached the edge of the Petri dish [50], using the following Formula (1):GII = [(MGC – MGT)/MGC] × 100 (1)
where MGC = mycelial growth in the control (mm) and MGT = mycelial growth under treatment (mm). The experiment included four repetitions and was replicated twice.

Dual cultures were used to evaluate antagonism against *B. exigua* var. *exigua*. For this, the three bacterial strains exhibiting the highest inhibition rates of pathogen growth were selected, while the strain CA10A was also used because it exhibited high antagonism against another pathogen that causes root rot in chicory, the oomycete *Phytophthora cryptogea* [1].

To determine GII, measurements of the mycelial growth of the pathogens were performed at 4, 7, 10, and 14 days of incubation by determining the distance between the antagonistic bacteria and the fungal hyphal edge. The experiment included three repetitions and was replicated twice. The two strains that showed the greatest inhibition of *B. exigua* var. *exigua* in dual cultures were selected for subsequent evaluations.

### 4.3. Evaluation of the Control Effectiveness of P. protegens on B. exigua var. exigua in Roots of Industrial Chicory Grown under Controlled Conditions

To determine the effectiveness of *P. protegens* on the control of *B. exigua* var. *exigua* in industrial chicory, roots were collected from plants grown at two experimental stations: Selva Negra (36°85′44″ S, 72°09′70″ W), San Ignacio, Ñuble Region, and Canteras (37°50′45″ S, 72°31′66″ W), Quilleco, Biobío Region, Chile. The samples were placed in clean containers and transported the same day to the Phytopathology Laboratory of the Faculty of Agronomy, University of Concepción, Chile. Upon arrival to the laboratory, the roots were carefully cleaned and washed with distilled water to remove soil residues and impurities and then stored in a chamber at 4 °C until use. 

To conduct the experiment, the chicory roots were disinfected with a 10% sodium hypochlorite solution for 10 min (*v*/*v*), then rinsed twice with sterile distilled water and subsequently dried in a laminar flow chamber [8,35]. Subsequently, a piece of tissue (10 mm in diameter) from the upper part of the root was extracted using a cork borer and then inoculated in that area with 300 µL of bacterial culture of strains ChB7 and Ca10A at a concentration of ≥1 × 10^6^ UFC mL^−1^, which was determined by using serial dilutions. After 48 h, a disc of actively growing mycelium (10 mm in diameter) of *B. exigua* var. *exigua* strain Pho669, previously grown in PDA medium at 25 °C for 7 days, was put on the wound inoculated with an antagonistic bacterium. Subsequently, the wounds were sealed with plastic wrap to prevent dehydration, and the roots were incubated in humid chambers at room temperature (±20 °C) for 21 days. The following treatments were evaluated: non-inoculated control, control inoculated with fungal strain Pho669, inoculation with bacterial strain Ca10A + fungal strain Pho669, and inoculation with bacterial strain ChB7 + fungal strain Pho669. On day 21, the roots were cut transversally in equal parts, and the disease damage index was determined by measuring the necrotic area (rot) on the root surface in a transversal and longitudinal way (mm) (disease damage index = (transversal + longitudinal damage)/2). The initial bacterial concentration in each trial was determined by a bacterial count. For this, cultures grown in King B at 25 ± 1 °C for 48 h were serially diluted seven times in sterile distilled water at a ratio of 1:10. Subsequently, a volume of 10 µL of each dilution was deposited on a Petri dish with King B (KB) agar. Once the microdrops were dry, the plates were incubated at 24 °C for 24 h, and a count of units forming colonies (CFU mL^−1^) was performed. 

The experiment was established using a completely randomized block design. Each block was represented by a humid chamber containing all the treatments under evaluation. The experiment was replicated twice and included four repetitions per treatment.

### 4.4. Evaluation of the Colonization Capacity of P. protegens in Industrial Chicory Roots Grown in Pots under Controlled Conditions

Ninety pre-germinated seeds of industrial chicory cv. Diesis were used. The seeds were placed in nursery trays of 50 alveoli (54.4 × 28.2 cm) with peat and perlite substrate (2:1) previously autoclaved (120 °C for 15 min) and kept in a growth room under controlled conditions (24 °C ± 1 °C) and a photoperiod of 14 h light/10 h darkness (day/night), delivered by far-red (630–660 nm), yellow (615 nm), blue (460–490 nm), and white light, with a spectrum of 68 to 88.3 µmol m^−2^ s^−1^.

Seven days after seeding (DAS), 30 chicory seedlings were inoculated with the bacterial strain Ca10A using a micropipette, 30 chicory seedlings were inoculated with the strain ChB7, and 30 chicory seedlings were used as controls (non-inoculated with the antagonistic bacteria). The initial concentration of the bacteria was adjusted to 1 × 10^6^ CFU mL^−1^, which was confirmed by bacterial count. Thirty days after bacterial inoculation, seedlings were transplanted to individual pots of 12 cm in diameter and 40 cm high, containing a substrate consisting of a mixture of peat and perlite (2:1), which was previously autoclaved at 120 °C for 15 min. The pots were kept in a growth room under the controlled conditions described above. The presence of bacteria in industrial chicory roots was evaluated at 30, 90, and 180 days after bacterial inoculation, using the *phlD*-specific PCR-based dilution end point assay method described by McSpadden-Gardener et al. [49]. Briefly, 1 g from the root surface was placed in a test tube with sterile distilled water for 24 h. Then, the liquid content of each tube was diluted four times (10^−1^, 10^−2^, 10^−3^, and 10^−4^) in ELISA microplates containing 150 µL of sterile water. Then, 50 µL of the dilutions 10^−2^, 10^−3^, and 10^−4^ were added to 150 µL of KB broth (1/3) with antibiotics, ampicillin (40 µg mL^−1^), chloramphenicol (13 µg mL^−1^), and cyclohexamide (100 µg mL^−1^), in microplates for a subsequent incubation at 25 °C for 48 h with constant agitation at 150 rpm. At the end of the incubation period, the presence of the *phlD* gene in *P. protegens* strains was determined directly from the bacterial cells present in the samples through the PCR technique by using the PCR parameters described by McSpadden-Gardener et al. [49]. The *phlD*+ samples were associated with the presence of the inoculated strains at the beginning of the trial. Measurements of the fresh and dry weights of roots and leaves (g) were performed at 180 days after bacterial inoculation.

The experiments were conducted using a completely randomized design, and each pot represented an experimental unit.

### 4.5. Evaluation of the Root Colonization Capacity and Control Effectiveness of Antagonistic Bacteria on Root Rot Caused by the Pathogen B. exigua var. exigua in Industrial Chicory Grown under Field Conditions

Two experiments were carried out at the Selva Negra (36°85′44″ S, 72°09′70″ W) and Canteras (37°50′45″ S, 72°31′66″ W) experimental stations, located in Ñuble and Biobío Regions, Chile, respectively. Both experimental stations were managed by the company Beneo Orafti S.A. At Selva Negra, the soil corresponds to an Andisol derived from volcanic ash, Pueblo Seco Series (associated with Arrayán Series); it was deep and well drained, with a loam-to-silt loam texture, flat topography with slopes from 0 to 1%, and organic matter content ranging from 8 to 12% in the A horizon, thus presenting high biological activity [36,51]. At Canteras, the soil belongs to an Arenales series, whose parent material is recent volcanic, andesitic-basaltic sand; it is characterized by a sandy-loamy to sandy texture with or without gravel, with depth of 30 and 80 cm and low content of organic matter, thus presenting moderate biological activity [36,51].

The climatic conditions observed at both experimental stations at the time of the experiment were obtained from the National Agroclimatic Network (AGROMET, Santiago, Chile) and are described in the Appendix A. At Selva Negra, rainfall reached 390 mm, being mainly concentrated in the months of September, October, and April; the average air temperature was warm, while the average soil temperature was quite stable, with minimum and maximum temperatures ranging from 13 to 17.6 °C during the season. Similar climatic conditions were observed at Canteras, with an accumulated rainfall of 294 mm in the growing season; the average air temperature was 2 °C higher than at Selva Negra, while soil temperatures were less stable with minimum and maximum temperatures ranging from 2.1 to 31 °C.

Before seeding, seeds of industrial chicory cv. Diesis were inoculated with the beneficial bacteria using the following methodology: four Falcon tubes containing a volume of 40 mL of bacterial culture in KB broth were centrifuged at 20 °C for 8 min at 4000 rpm, and the supernatant was removed. The bacterial pellet was washed and resuspended in 25 mL of saline solution (0.9% NaCl) and centrifuged at 20 °C for 5 min at 4000 rpm. The supernatant was discarded, and 100 µL of sterile King B broth and 3 mL of carboxymethylcellulose (0.05%) were added to the pellet. The mixture was slightly vortexed, then added to the seed in order to obtain an inoculation dose of 7.5 mL of each concentrated microorganism per 100 g of chicory seeds, and finally left to dry at room temperature (~23 °C) for 18 h.

The seeds were sown on 17 September and 8 October 2015 at Selva Negra and Canteras, respectively. At both experimental sites, a six-row precision planter (MS high-precision model, Monosem, Paris, France) was used. The experimental units consisted of plots of 3 m × 7 m with six rows. The planting spacing was 10.2 cm between plants and 45 cm between rows. The treatments were established in a randomized complete block design with four repetitions. The evaluated treatments are described in Table 4.

The agronomic management of the experiments (weed control, irrigation, and fertilization) was conducted according to the standard management recommended by Beneo Orafti S.A. Fertilization was applied at seeding and consisted of a mixture of 30 units of N, 100 units of P_2_O_5_, 250 units of K_2_O, 2 units of B, 1.8 U Zn, 24 units of Mg, and 32 units of S ha^−1^. An accumulated irrigation of 380 mm was applied by fixed spray sprinklers. Leaf pest control was carried out using an insecticide based on Thiamethoxam (141 g L^−1^) and Lambda-cyhalothrin (106 g L^−1^) (Engeo^®^ 247 ZC, Syngenta) applied at a dose of 150 mL ha^−1^ when plants reached the growth stage of three to four true leaves (45 DAS). For leaf disease control, two fungicide applications based on Fenpropidin (375 g L^−1^) and Difeconazole (100 g L^−1^) (Score Beta^®^ 475 EC, Syngenta] were used in the growth stage of 20–25 true leaves (120 and 150 DAS). Weed control was carried out using standard herbicides; at pre-seeding, Trifluralin (48 g L^−1^) (Treflan^®^ EC, Agrotechnology) was applied at 1.5–2.0 L ha^−1^; in post-emergence (cotyledon stage), Propyzamide (50% *w*/*w*. 500 g kg^−1^) (Kerb^®^ 50W, Dow AgroSciences LLC) was applied at a dose of 0.5 kg ha^−1^, and Flumetsulam (80%) (Preside^®^ 80WG, Dow AgroSciences LLC.) was applied at a dose of 10–20 g ha^−1^ at the growth stage of 2–2.5 true leaves, Imazamox (700 g kg^−1^) (Sweeper^®^ 700 DG, Basf Chile S.A.) was applied at a dose of 20 g ha^−1^ at the stage of three to four true leaves, S-metallochlor (960 g L^−1^) (Dual^®^ Gold 960 EC, Syngenta) was applied at a dose of 200 mL ha^−1^ and Dimethenamide-P (720 g L^−1^) (Frontier^®^-P EC, Basf Chile S.A.) was applied at 200 mL ha^−1^, and at post-emergence, Lenocil (800 g kg^−1^) (Venzar^®^ WP, DuPont Chile S.A.) was applied at a dose of 500 g ha^−1^. 

The inoculation of *B. exigua* var *exigua* was carried out in the four central rows of the experimental plots. An amount of 10 g of inoculum was distributed manually per linear meter in a furrow 8–10 cm deep at the side of the row. This was conducted at the growth stage of 9–10 true leaves between 8 and 10 December 2015 (approximately 80 DAS). For the assays under field conditions, the preparation of the artificial inoculum of the strain Pho669 consisted of multiplying the pathogen for 30 days in 500 mL Erlenmeyer flasks with previously sterilized millet grains. For this, each flask was filled with 200 g of millet, 100 mL of distilled water, and 30 discs (7 mm in diameter) of active growth of the fungus [4,52,53].

The trials were harvested manually between 16 and 17 May 2016 (about 8 months after seeding). The weight of healthy and diseased roots (ton ha^−1^), as well as the fresh leaf weight (ton ha^−1^), were measured using a digital platform scale (model DY61, Maigas Comercial S.A., Santiago, Chile) in the field. Additionally, colonization by *P. protegens* bacteria was determined during different growth stages of the chicory. For this, approximately 10 plants were obtained from each experimental plot at 30, 60, 90, and 150 DAS and harvest, including the treatments without bacterial inoculation. The presence or absence of the antagonistic bacterial strains in the rhizosphere of the chicory plants was determined through the amplification of the *phlD* gene by means of PCR following the protocols described previously [49]. In chicory, the root growth rate increased after 100 DAS, and thus, the roots thickened considerably with accumulating food reserves for flowering. The presence of bacteria in the upper and lower parts of the root was evaluated at 150 DAS and harvest; a sample was considered positive for the *phlD* gene when the upper and/or lower parts of the root were positive for the presence of this gene.

Sugar beet roots were individually rated for rot using a modified seven-level scale used by Beneo Orafti S.A., where 0 = no rot symptoms, 1 = with surface rot spots, 2 = with rot band of <5 mm penetration, 3.1 = with less than 5 mm penetration into the rot band and <30% root rot, 3.2 = with more than 5 mm of penetration in the rot band and >30% rot, 4.1 = root tip rot and advance <5 cm from apex, 4.2 = advanced tip rot >5 cm advance from apex, and 5 = 100% completely rotten root (see Appendix A). The evaluations were carried out in the field, once the four central rows of each plot were harvested. The obtained data were used to determine the incidence of the disease, which corresponded to the total number of roots with rot symptoms of the total number of roots harvested. The disease severity was evaluated through the following modified Formula (2) of [54]: (2)$$\mathrm{Severity}\;\left(\%\right)=\frac{\left(0\times 0E\right)+\left(1\times 1E\right)+(2\times 2E)+(3\times 3.1E)+(4\times 3.2E)+(5\times 4.1E)+(6\times 4.2E)+(7\times 5E)}{N\times G}\times 100$$
where E is the level on the root rot scale, N is the total number of roots, and G is the maximum severity in the rot scale.

### 4.6. Statistical Analysis

The preselection was evaluated by using Ward’s hierarchical clustering method, while average linkage was used for clustering. Data from each conglomerate or cluster were analyzed for normality using the Shapiro–Wilk test. An analysis of variance (ANOVA) was used to evaluate the results of the dual culture experiment, while significant differences between the treatments (*p* ≤ 0.05) were analyzed by Tukey’s HSD test (α = 0.05). The data obtained from the experiment in humid chambers were evaluated using the non-parametric Kruskal–Wallis test, while pairwise comparisons were used to determine differences between the treatments. Data on the biomass obtained from the bacterial colonization test in the growth chamber and on the yield obtained in the field tests were analyzed for normality using the Shapiro–Wilk test. An analysis of variance (ANOVA) was used to evaluate the experiments in the growth chamber and in the field according to the experimental design, while significant differences between the treatments (*p* ≤ 0.05) were analyzed by Fisher’s LSD test (α = 0.05). The results (as percentages) were transformed using the Bliss angular transformation prior to the statistical analysis. All statistical analyses were performed using SAS (Statistical Analysis System) software version 8.0 [55].

## 5. Conclusions

This study is the first report of the inhibitory effect exerted by *P. protegens* bacteria on the fungus *B. exigua* var. *exigua* affecting roots of industrial chicory. Bacterial strains ChB7and Ca10A used as seed treatments were capable of surviving in the chicory rhizosphere during crop development, which would indicate that root exudates released by industrial chicory plants favored the survival of *P. protegens*. Higher percentages of bacterial samples positive for the *phID* gene were observed at Selva Negra compared with Canteras, which could be explained by the differences in soil type and soil properties between the experimental stations. *P. protegens* strains ChB7and Ca10A selected in vitro and under controlled conditions were able to colonize roots, protect plants from *B. exigua* attack, and increase plant productivity between 21 and 24% with respect to the inoculated control. In conclusion, strains of *P. protegens* inhibited the mycelial growth of the fungus *B. exigua* var. *exigua* under in vitro conditions and reduced root rot levels under in vivo and field conditions in the cultivation of industrial chicory in two ecosystems of Chile.

## Figures and Tables

**Figure 1 plants-13-00263-f001:**
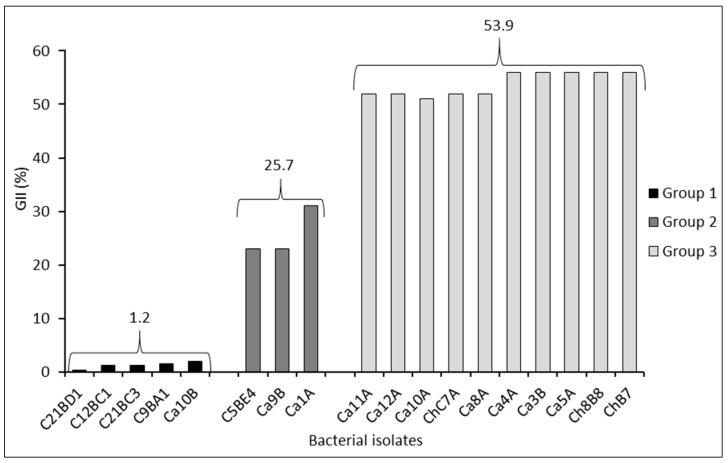
Grouping of bacterial isolates by mycelial growth inhibition index (GII) of *Boeremia exigua* var. *exigua* strain Pho669 after 14 days of incubation under in vitro conditions in PDA medium determined by Ward’s cluster analysis. Numbers of the same color over the parentheses enclosing the group of columns correspond to the average (%) between the strains of each group. Clustering or grouping was conducted by using average linkage with the Ward’s hierarchical clustering method.

**Figure 2 plants-13-00263-f002:**
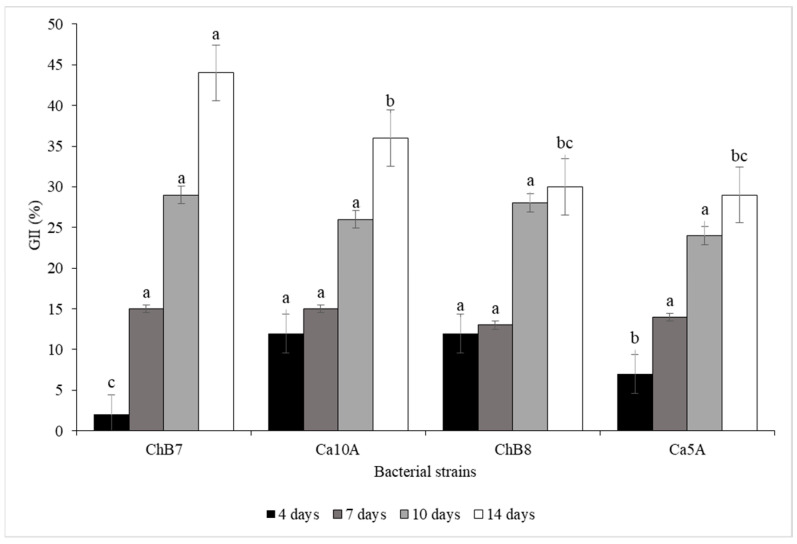
Mycelial growth inhibition index (GII) of *Boeremia exigua* var. *exigua* strain Pho669 obtained in dual cultures by four strains of *Pseudomonas protegens* bacteria over time, under in vitro conditions. Bars indicate standard error for each date of assessment. Different letters on bars of the same color indicate significant difference by Tukey’s HSD test (α = 0.05).

**Figure 3 plants-13-00263-f003:**
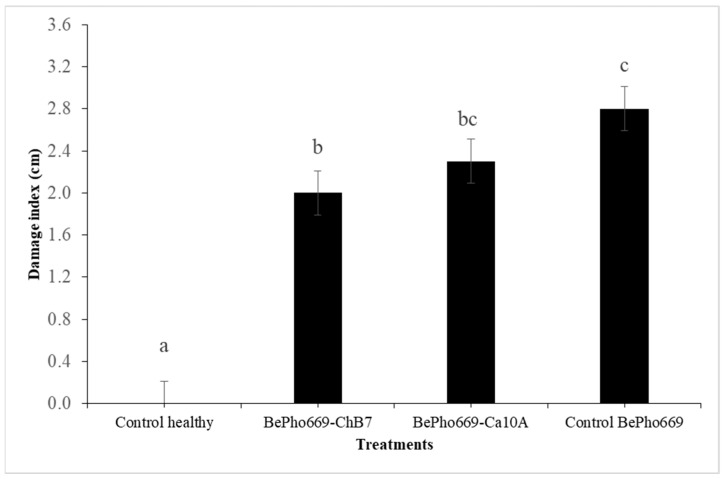
Damage index (cm) of root rot obtained with strains ChB7 and Ca10A of *Pseudomonas protegens* on *Boeremia exigua* var. *exigua* strain Pho669 (BePho669) after 21 days of incubation on inoculated industrial chicory roots in a humid chamber. Data are expressed as mean ± standard error of eight repetitions. Different letters in the columns indicate significant differences between treatments according to the non-parametric Kruskal–Wallis test (*p* ≤ 0.05). Bars indicate the standard error.

**Figure 4 plants-13-00263-f004:**
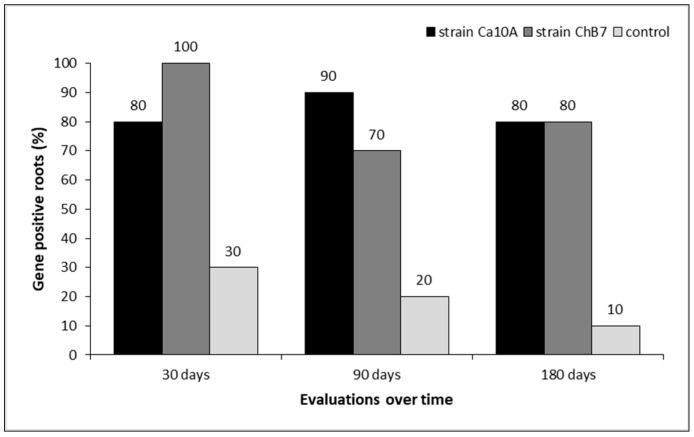
Results of industrial chicory root samples positive for the presence of the *phID* gene (%), inoculated with two strains of *Pseudomonas protegens* and without bacterial inoculation (control) in an experiment under controlled conditions. Results based on three evaluation dates after bacterial inoculation. Percentage (%) calculated based on 10 roots per treatment in each date of assessment.

**Figure 5 plants-13-00263-f005:**
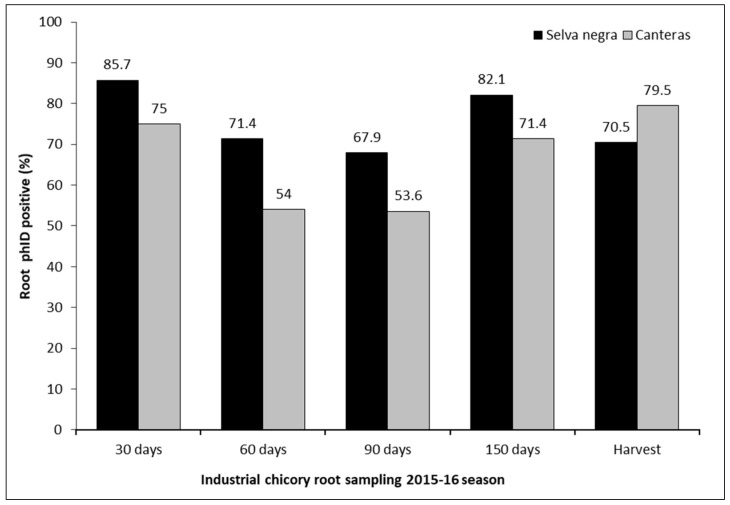
Presence of the *phID* gene in bacterial populations obtained from industrial chicory roots that were seed-inoculated with *Pseudomonas protegens*, sampled at 30, 60, 90, and 150 DAS and harvested at the Selva Negra and Canteras experimental stations (Beneo-Orafti S.A), Ñuble and Biobío Region, Chile, respectively.

**Table 1 plants-13-00263-t001:** Yield in ton ha^−1^ (leaf weight and healthy and diseased roots) obtained with inoculation of the pathogenic *B. exigua* var. *exigua* strain Pho669 and antagonistic bacteria *P. protegens* at harvest under field conditions. Experiment conducted at the Selva Negra experimental stations, Biobío Region. Different letters in the same column for the variable assessed show differences between treatments means according Fisher’s LSD test (α = 0.05).

Treatments	Fresh Leaf Weight (ton ha^−1^)		Healthy Root Weight (ton ha^−1^)		Diseased Root Weight (ton ha^−1^)	
Non-inoculated control	16.5 ± 0.6	b	98.6 ± 3.7	a	11.7 ± 1.1	ab
Control inoculated with Pho669	13.5 ± 1.0	a	93.4 ± 2.9	a	18.5 ± 2.9	c
Seed inoculated with ChB7	14.8 ± 0.6	ab	97.8 ± 1.7	a	10.1 ± 1.2	a
Seed inoculated with ChB7 + Pho669	14.0 ± 0.5	a	100.5 ± 0.9	ab	11.1± 0.2	ab
Seed inoculated with Ca10A	16.2 ± 0.3	b	105.2 ± 2.1	ab	10,3 ± 0.9	a
Seed inoculated with Ca10A + Pho669	13.0 ± 0.6	a	102.8 ± 2.8	ab	7.6 ± 0.3	a
Seed inoculated with Ca10A + ChB7	14.4 ± 0.4	a	111.9 ± 3.0	b	9.7 ± 1.7	a
Seed inoculated with Ca10A + ChB7 + Pho669	15.6 ± 0.6	ab	95.9 ± 0.7	a	17.6 ± 1.2	c
Coefficient of variation (CV) %	10.2		6.9		33.4	
*p*-value	0.041		0.021		0.005	

**Table 2 plants-13-00263-t002:** Yield in ton ha^−1^ (leaf weight and healthy and diseased roots) obtained with inoculation of the pathogenic *B. exigua* var. *exigua* strain Pho669 and antagonistic bacteria *P. protegens* at harvest under field conditions. Experiment conducted at the Canteras experimental station, Biobío Region. Different letters in the same column for the variable assessed show differences between treatments means according Fisher’s LSD test (α = 0.05).

Treatments	Fresh Leaf Weight (ton ha^−1^)		Healthy Root Weight (ton ha^−1^)		Diseased Root Weight (ton ha^−1^)	
Non-inoculated control	10.3 ± 0.7	a	59.6 ± 1.8	ab	3.3 ± 1.0	a
Control inoculated with Pho669	11.5 ± 0.8	ab	53.5 ± 1.7	a	5.5 ± 0.7	ab
Seed inoculated with ChB7	11.5 ± 0.4	ab	64.8 ± 1.4	b	2.7 ± 0.2	a
Seed inoculated with ChB7 + Pho669	11.9 ± 0.6	ab	61.5 ± 1.2	ab	9.4 ± 0.9	b
Seed inoculated with Ca10A	10.7 ± 0.6	ab	61.3 ± 1.5	ab	2.0 ± 0.2	a
Seed inoculated with Ca10A + Pho669	11.5 ± 0.6	ab	62.6 ± 1.8	ab	2.0 ± 0.2	a
Seed inoculated with Ca10A + ChB7	12.7 ± 0.5	b	66.2 ± 1.3	b	4.3 ± 0.5	ab
Seed inoculated with Ca10A + ChB7+ Pho669	10.1 ± 0.7	a	60.5 ± 1.8	ab	4.3 ± 1.3	a
Coefficient of variation (CV) %	12.5		10.8		42.5	
*p*-value	0.001		0.043		0.004	

**Table 3 plants-13-00263-t003:** Incidence and severity of the root rot disease (%) obtained with inoculations of *B. exigua* var. *exigua* strain Pho669 and strains of *P. protegens* at harvest under field conditions. Experiment conducted at the Selva Negra and Canteras experimental stations (Beneo-Orafti S.A), Ñuble and Biobío Regions, Chile. Different letters in the same column for the variable assessed show differences between treatments means according Fisher’s LSD test (α = 0.05).

Treatments	Root Rot Incidence (%)				Root Rot Severity (%)			
	Selva Negra		Canteras		Selva Negra		Canteras	
Non-inoculated control	62.7 ± 1.4	a	16.2 ± 2.3	a	25.8 ± 0.3	a	7.7 ± 1.5	a
Control inoculated with Pho669	81.1 ± 5.6	b	22.9 ± 1.9	a	35.6 ± 1.7	b	11.9 ± 1.3	a
Seed inoculated with ChB7	56.9 ± 1.6	a	20.3 ± 1.3	a	28.0 ± 1.4	ab	9.6 ± 0.6	a
Seed inoculated with ChB7 + Pho669	66.8 ± 0.2	a	43.1 ± 4.3	b	25.9 ± 0.2	a	23.5 ± 3.4	b
Seed inoculated with Ca10A	61.0 ± 2.8	a	22.0 ± 1.6	a	23.4 ± 1.0	a	9.4 ± 0.7	a
Seed inoculated with Ca10A+ Pho669	62.7 ± 2.2	a	23.5 ± 3.7	a	23.8 ± 1.0	a	10.6 ± 1.7	a
Seed inoculated with Ca10A + ChB7	58.3 ± 2.2	a	24.9 ± 2.2	ab	23.4 ± 0.8	a	11.7 ± 1.1	a
Seed inoculated with Ca10A + ChB7 + Pho669	66.9 ± 1.0	a	29.9 ± 4.6	ab	27.8 ± 0.5	ab	13.1 ± 2.3	ab
Coefficient of variation (CV) %	12.5		26.3		8.7		28.9	
*p*-value	0.033		0.023		0.002		0.041	

**Table 4 plants-13-00263-t004:** Treatments used to determine the capacity of *Pseudomonas protegens* bacteria to colonize industrial chicory roots and control root rot caused by the pathogen *Boeremia exigua* var. *exigua* strain Pho669. Experiment conducted at the Selva Negra and Canteras experimental stations (Beneo-Orafti S.A), Ñuble and Biobío Regions, Chile.

Treatment	Description
T1	Non-inoculated control
T2	Control inoculated with Pho669
T3	Seed inoculated with strain ChB7
T4	Seed inoculated with strains ChB7 and Pho669
T5	Seed inoculated with strain Ca10A
T6	Seed inoculated with strains Ca10A and Pho669
T7	Seed inoculated with strains Ca10A and ChB7
T8	Seed inoculated with strains Ca10A and ChB7, and Pho669

## Data Availability

The data presented in this study are available on request from the corresponding author.

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
