# Peer review of "Use of Pseudomonas protegens to Control Root Rot Disease Caused by Boeremia exigua var. exigua in Industrial Chicory (Cichorium intybus var. sativum Bisch.)"

_plants, 2024, doi:10.3390/plants13020263_

Round 1

Reviewer 1 Report

Comments and Suggestions for Authors

The submitted research paper “  Use of Pseudomonas protegens to control root rot disease caused 2 by Boeremia exigua var. exigua in industrial chicory (Cichorium 3 intybus var. sativum Bisch.)” is very interesting to read and contributed greatly in this field.

I have few comments to improve this manuscript

In the introduction section, please update latest references to reduce the research gap.

The following two statements required more explanation and add adequate references to strength the statements.

“In industrial chicory, B. exigua var. exigua causes dark brown to black ne-44 crotic lesions with a dry and firm consistency, mainly located in the upper part of the roots 45 [4]. Currently, there is no chemical or varietal control of this pathogen in industrial chicory 46 [9],”

Explain the following clearly. How do you determine the fungus as highly pathogenic? Please explain the statement carefully.

4.1. Isolates of B. exigua var. exigua 349

The highly pathogenic fungus B. exigua var. exigua strain Pho669, which was obtained 350 from diseased roots of industrial chicory, was used in all the experiments.

Please do adequate changes and present the following. Remove unpublished or personal communication data from this manuscript.

If possible, please provides images to support the data on “Disease damage index”

roots (Test-Student= 0.71 and P=0.2078)., here, 0.71 is what value? F-value?

Non-inoculated control

16.5 ± 0.6

b

98.6 ± 3.7

a

11.7 ± 1.1

ab

To express variations, use superscripts., eg., 16.5 ± 0.6b

Please check throughout the manuscript.

Please revise conclusion with significant findings and recommendation.

Author Response

We really appreciate if the reviewer can review the attached file and considering to review an improved version of our manuscript.

Reviewer 2 Report

Comments and Suggestions for Authors

The ms plants-2732219 with the title of Use of Pseudomonas protegens to control root rot disease caused by Boeremia exigua var. exigua in industrial chicory (Cichorium intybus var. sativum Bisch.) investigates an important topic but the authors should improve their ms.

L11: there is no chemical or varietal control for this disease, are you sure?

L18 (P≤0.05),  Pshould be italic. Correct this in whole ms

L55-59 Interest in the biological control of crop diseases has increased in recent years due to increasing public concern about the impact of chemical pesticides on the environment, human health, and food safety [10,11]. In this sense, the use of microorganisms capable of controlling soil pathogens constitutes an environmentally friendly alternative for the control of industrial chicory root rot>> Please use these citations for this text “Phylogenetic Diversity of Trichoderma Strains and Their Antagonistic Potential against Soil-Borne Pathogens under Stress Conditions”       “Biological Control of Celery Powdery Mildew Disease Caused by Erysiphe heraclei DC In Vitro and In Vivo Conditions”          “Can symbiotic bacteria (Xenorhabdus and Photorhabdus) be more efficient than their entomopathogenic nematodes against Pieris rapae and Pentodon algerinus Larvae?”     “New Bacillus subtilis Strains Isolated from Prosopis glandulosa Rhizosphere for Suppressing Fusarium Spp. and Enhancing Growth of Gossypium hirsutum L.”

L76-80 please revise the aims at end of the introduction.

Please define all abbreviations in first mention, check this

Can you add the SE or any statistical analysis for Figure 1?

L113 it is not correct to say (P = 0.001), but you can say (P 0.001)

Figure 3: ± standard error of 8 repetitions is completely wrong, I think the authors made a mistake when they wanted to attach the ± standard error above the columns, please recheck and revise it because see how big is ± standard error for control or other treatments.

L148 if there is not significant, then remove (P ≥ 0.05)

Table 2: Leaf weight as fresh or dry?

Discussion section should be improved and the authors should cite uncited text. Please remove the common text.

L459 why authors used this deign? The experiments were conducted using a completely randomized design and each pot represented an experimental unit.

Why authors did not use A completely randomized design CRD?

Conclusion, please revise it and write the most important findings and add some values

References, use some recent publications.

Comments on the Quality of English Language

Some edits are needed

Author Response

We really appreciate if the reviewer can review the attached file and considering to review an improved version of our manuscript 

Reviewer 3 Report

Comments and Suggestions for Authors

Manuscript review: "Use of Pseudomonas protegens to control root rot disease caused by Boeremia exigua var. exigua in industrial chicory (Cichorium intybus var. sativum Bisch.)" by Tamara Quezada-D'Angelo at al.

I find the research interesting; however, the experiments were not carried out correctly. It is unclear how the control plants, which were free of infection, inoculated with BCA bacteria, and infected with Boeremia exigua, showed a similar level of infection. Furthermore, the study did not test soil and seeds for the presence of Boeremia exigua. The research could be improved by using qPCR techniques to assess the level of infection and inoculation of BCA plants. Additionally, the authors should increase the number of plants in the sample size to get a more representative result. The level of infection (Damage index) was performed on eight plants, which is not a representative research sample. I believe that the research has potential and the authors should continue by eliminating the methodological errors made. The current version is not suitable for printing.

Here are the specific corrections and suggestions for improvement:

1. L14: Clarify the method used in the study.

2. L26: Use "chicory" as a keyword instead of repeating the information from the title.

3. L79-80: Specify the method used in this part of the study.

4. L82- Pseudomonas protegens isolates?

5. Include photographic material documenting the experiment in the chapter that describes in vitro preselection of the antagonistic.

6. Plot the results of the statistical analysis on a graph in figure 2.  What is an error standard?

7. L113. On which sample of plants was the study carried out? Give the value (N).

8. Clarify the title of the diagram in L122 by stating the experimental conditions (laminar chamber). Additionally, clarify if the study was conducted on 8 plants and provide more details.

9. L126. Error standard?

10. L142. What method was used to assess the presence of the phID gene? What was the internal control? For what reason was the qPCR technique not used? What were the DNA parameters? Please present the results in the form of gels.

11. L157. Test-Student= 0.71.... Please include a reference in the working methodology.

12. Table 1,2. give the N value for each parameter. In how many replicates was the study conducted. How were Healthy and Diseased roots separated for root weight calculation? How was it confirmed which species of fungus was the culprit of disease symptoms? Were mycological analyses conducted to confirm the effectiveness of the infection? Leaf weight was given to healthy or diseased plants? Tabble 1. P-value = 0005?

13 Table 3. Which fungus was the culprit of Root rot in the control? How do the authors explain the results obtained? For the location of the experiment in Canteras, it was found that Seed inoculated with ChB7 + Pho669 caused an increase in Root rot incidence (%) and Root rot severity compared to the control objects. Was the infection caused solely by B. exigua? What was the presence of other pathogens on the roots? How do the authors explain the occurrence of Root rot at an identical statistical level in Non-inoculated control and Control inoculated with Pho669 and Seed inoculated with ChB7 combinations? What is the P-valor?

14. L350. Where is the isolate stored? Give the name of the collection?

15. L355-359. Where are the isolates stored? When and where were they isolated?

16. L387. Phytophthora cryptogea is not a pseudo-fungus. It belongs to Oomycetes and Chromista.

17. L410.,441. How was the concentration of f 1x106 UFC mL-1 calculated?

18. L417-419. Who is the author of the method? Cite the literature. On how many plants were the studies conducted?

19. L455. Describe the details of the PCR technique. Was the DNA isolated first? By what method? What were the parameters?

20. Tables 4 and 5 Transfer to supplement.

21. L540. Give the parameters of the inoculum. How was the efficacy of infection assessed? Was the soil tested for the presence of Boeremia exigua var. exigua? Was the seed used for sowing free of Boeremia exigua. What was the severity of other diseases on the plants?

22. L568-570. On which sample of plants was e incidence of the disease and Disease severity assessed?

23. Supplementary material needs to be corrected and translated into English. It should also be clarified which parameters it refers to. In addition, the spelling should be standardised by using Boeremia instead of Phoma.

Author Response

(The authors gave the same response as above.)

Round 2

Reviewer 2 Report

Comments and Suggestions for Authors

The Ms was improved, thanks authors 

Comments on the Quality of English Language

Just some minor edits are needed 

Reviewer 3 Report

Comments and Suggestions for Authors

The manuscript has been revised. After reviewing it again, I have no comments and recommend that the paper be accepted for publication in its current form. I thank you for all the clarifications and wish you good luck with the realization of further studies.